# Pick and place process for uniform shrinking of 3D printed micro- and nano-architected materials

Tomohiro Mori [1,2] ✉, Hao Wang [1,3,4] ✉, Wang Zhang [1], Chern Chia Ser[1], Deepshikha Arora[1], Cheng-Feng Pan[1,5], Hao Li[1], Jiabin Niu[1], M. A. Rahman [1], Takeshi Mori[2], Hideyuki Koishi[2] & Joel K. W. Yang [1] ✉

Two-photon polymerization lithography is promising for producing three-dimensional structures with user-defined micro- and nanoscale features. Additionally, shrinkage by thermolysis can readily shorten the lattice constant of three-dimensional photonic crystals and enhance their resolution and mechanical properties; however, this technique suffers from non-uniform shrinkage owing to substrate pinning during heating. Here, we develop a simple method using poly(vinyl alcohol)-assisted uniform shrinking of three-dimensional printed structures. Microscopic three-dimensional printed objects are picked and placed onto a receiving substrate, followed by heating to induce shrinkage. We show the successful uniform heat-shrinking of three-dimensional prints with various shapes and sizes, without sacrificial support structures, and observe that the surface properties of the receiving substrate are important factors for uniform shrinking. Moreover, we print a three-dimensional mascot model that is then uniformly shrunk, producing vivid colors from colorless woodpile photonic crystals. The proposed method has significant potential for application in mechanics, optics, and photonics.

Among additive manufacturing technologies, two-photon polymerization lithography (TPL) is one of the most promising for producing micro- and nanoscale features[1,2], and has greatly facilitated research in mechanics[3–5], microfluidics[6,7], micro robots[8,9], and biology[10–13]. In particular, given the intrinsic optical transmittance of polymers, the technique is suited for optics and photonics applications, such as structural color printing[14–17], integration of lenses at the end of optical fibers[18,19], optical elements[20–23], and photonic crystals[24–27]. However, the resolution of TPL is limited by the structural rigidity of the material, diffraction limit, and proximity effects during exposure[28], hampering the printing of sub-wavelength structures necessary for manipulating visible light. While stimulated emission depletion direct laser writing and diffusion-assisted high-resolution

TPL approaches have been proposed to improve the resolution[29–31], these methods require customized photoresists and complex systems. Alternatively, heat-shrinking by thermolysis can help us directly obtain nanoscale features with high resolution and accuracy, as well as enhance the mechanical properties of the structure[32–34]. In our previous work, we reported three-dimensional (3D) printed photonic crystal structures with a stopband and slow light modes in the visible spectrum using a heat-shrinking process[35]. The woodpile photonic crystal structures were fabricated using commercial equipment (Nanoscribe GmbH, Photonic Professional GT) and a widely used commercial IP-Dip resin. Various structural colors were achieved, indicating that the lattice constant was controlled from 300 to 650 nm by heat-shrinking. In addition, Zhang et al. fabricated nanolattices

[1]Engineering Product Development, Singapore University of Technology and Design, Singapore 487372, Singapore. [2]Industrial Technology Center of Wakayama Prefecture, Wakayama 6496261, Japan. [3]College of Mechanical and Vehicle Engineering, Hunan University, Changsha 410082, China. [4]Greater Bay Area Institute for Innovation, Hunan University, Guangzhou 511300, China. [5]Department of Electrical and Computer Engineering, National University of Singapore, Singapore 117576, Singapore. ✉e-mail: tomohiro_mori@wakayama-kg.jp; whchn@live.cn; joel_yang@sutd.edu.sg

using IP-Dip photoresist, followed by pyrolysis at 900 °C[36]. These pyrolytic carbon structures achieved densities below 1.0 g/cm³ and GPa-level strengths, indicating excellent lightweight and robust properties.

In these thermolysis studies, the base of the structure is easily deformed during the heating process, as it turns into a net-like mesh from shrinking because of the adhesion between the bottom layer and the substrate (marked with red in the scanning electron microscopic (SEM) image in Fig. 1a). Deformation of the structure is undesirable for practical applications as it causes non-uniform mechanical and optical properties. Sacrificial support structures, such as springs, scaffolds, and pedestals, have been used to support the main structures and partially circumvent this issue[24,27,37–39]. Further, the height of the designed structure can be increased to obtain a relatively uniform structure at the top, at the expense of sacrificing the distorted base and increased printing time[35]. Thus far, no approach can result in uniform shrinkage of the entire 3D print. Furthermore, 3D printed structures will be much more useful if they can be transferred from the substrate used during TPL printing onto target substrates, perhaps for integration with other devices.

In this study, we developed a simple method using poly(vinyl alcohol) (PVA) as a functional layer to achieve uniform shrinking of 3D printed structures. The 3D printed structures were detached from the PVA layer on the original substrate and transferred onto a receiving substrate to form a weak interaction between the base of the printed structures and receiving substrate. PVA, a water-soluble polymer, can survive the development process using organic solvents, but it easily dissolves in water, which means that the 3D structures can be readily detached from the original substrate. Subsequently, the 3D structures on the receiving substrate were heated and shrunk to obtain a uniformly scaled version of the original shape. The functional layer of PVA is an advantage of this proposed method for mitigating the adhesion between the structure and substrate. Structures with various micro- and nanoscale features were fabricated and uniform shrinking was observed in optical and electron micrographs to demonstrate the capability of this process. The surface roughness, surface free energy, and adhesion force of the prepared structures were measured to understand the uniform shrinking mechanism. Finally, we achieved a colorful and undistorted 3D mascot model composed of woodpile photonic crystals without sacrificial support structures. This technique can be readily applied to fabricate 3D structures with high mechanical stability consisting of nanoscale features, as it easily controls the uniform shrinking of 3D structures in any shape, size, position, and direction. The pick and place process further enables integration of 3D printed optical components with relevant devices on the recipient substrate.

## Results

3D uniform shrinking structures were prepared using the simple process illustrated in Fig. 1b (details in Methods). The 3D objects were first printed on top of a spin-coated layer of PVA of 90–100 nm thickness using the acrylic resin IP-Dip2 and TPL method, before removing the unexposed photoresist using a standard development procedure (hereafter referred to as as-printed, Fig. 1c). The PVA layer was dissolved in water, and the structures were collected using a hydrophilic membrane filter. Afterwards, the structures were manually transferred to the receiving substrate with microneedles. The position and direction of the structure could be easily controlled using our custom-made microneedle pickup system (Supplementary Fig. 1). Subsequently, the structures were heated in air at a rate of 10 °C/min from room temperature (RT) to 450 °C and held at this temperature for 5 min (hereafter referred to as post-processed, Fig. 1f). For comparison, the same 3D structures were also printed directly on a fused silica substrate and heated (without transfer to the

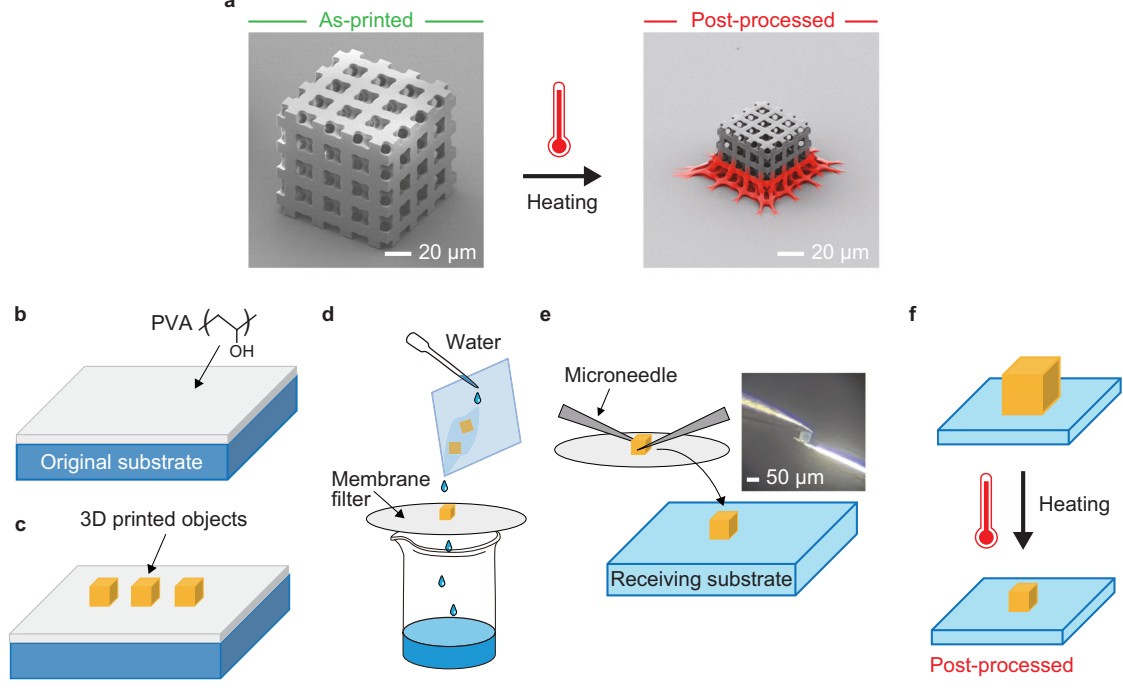

**Fig. 1 | Concept and schematic of the fabrication process. a** The problem in conventional direct heating method that results in the undesirable distortion of the region pinned to the substrate (indicated in red). Tilted-view SEM images of as-printed (before shrinking) and post-processed (after shrinking) structures. Schematic of the proposed fabrication process. **b** Spin-coating of PVA on the original substrate. **c** 3D printing on top of the PVA layer. **d** Dissolving the PVA layer and collecting the 3D structures with a membrane filter. **e** Transferring of the 3D structures to receiving substrate. Optical micrograph (inset) shows the top view of the transfer of the 3D structure using two microneedles. **f** Heating of sample to achieve uniform shrinkage.

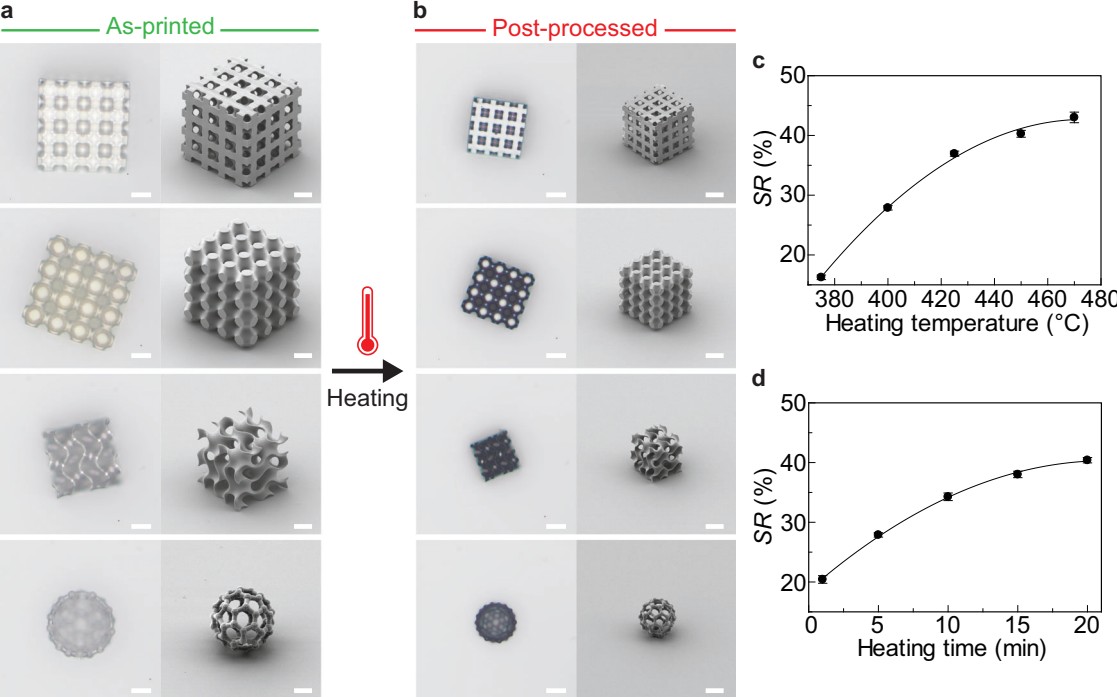

**Fig. 2 | Uniform shrinkage of printed 3D model with microscale features. a** Top view optical micrographs with bright-field reflective illumination and side view SEM images of the as-printed 3D structures. **b** Corresponding optical and SEM images of post-processed 3D structures. **c** Shrinkage rate of the 3D model (large woodpile structure) at different heating temperatures (heating time: 5 min, laser power: 27.5 mW). **d** Shrinkage rate of large woodpile structure for different heating times (heating temperature: 400 °C, laser power: 27.5 mW). Scale bars in **a**, **b** = 20 μm. The error bars in **c**, **d** represent standard deviations.

receiving substrate) under the same conditions as those used in the proposed method.

First, we investigated the ability of the proposed process flow to produce uniformly shrunk structures. Figure 2 shows the top (optical) and side (SEM) views of the as-printed and post-processed 3D structures of different shapes and sizes (with feature sizes of several micrometers). All of the structures in Fig. 2 were fabricated at a laser power of 27.5–30 mW to enable them to form with minimal shrinkage during the development process using an organic solvent. They were then transferred to a fused silica receiving substrate using microneedles and heated up to and maintained at 450 °C for 5 min. As a result, they successfully shrank uniformly without distortion compared with the conventional direct heating method (Fig. 1a). The shrinkage rate (SR) was defined by the following equation:

$$SR\,(\%) = (S_a - S_p)/S_a \times 100 \tag{1}$$

where $S_a$ is the size of the as-printed structure and $S_p$ is the size of the post-processed structure. The successful shrinkage rate in our method was $SR = 37$–42% for different structures, with the same degree of shrinkage along each axis of the structure (Fig. 2a, b and Supplementary Fig. 2). In addition, the SRs for the 3D printed structures with different faces remounted on the receiving substrate were mostly the same (Supplementary Fig. 3). Shrinkage originates from material decomposition through a thermo-oxidative degradation mechanism. Through thermogravimetric analysis (TGA) under an air atmosphere (Supplementary Fig. 4a), we first observed a small reduction in the mass of the IP-Dip2 acrylic photoresist, which is associated with the loss of adsorbed and absorbed water. A large weight loss was observed at 350–450 °C, which was associated with carbonization, followed by a smaller peak at 540 °C, which was associated with the decomposition of polymeric chains[40–42]. The IP-Dip2 photoresist is composed mainly of pentaerythritol triacrylate, which contains a large fraction of bonded oxygen. During the heating process, the formation of volatile

species such as CO and $CO_2$ by carbonization causes a large volume loss. We also analyzed the Raman spectra of the photoresist before and after heating (Supplementary Fig. 5). Thermolysis reduced the intensities of the peaks associated with the C–H (2947 cm⁻¹), C=O (1722 cm⁻¹), and C–O (935 cm⁻¹) stretching modes. By contrast, peaks corresponding to $sp^2$-rich carbon (2500–3100 cm⁻¹), graphitic carbon (1593 cm⁻¹), and disordered carbon (1353 cm⁻¹) were observed[43,44]. These results indicate that the photoresist formed an activated carbon-like structure with increased carbon content. The heating temperature range of 390–450 °C corresponded to drastic changes in volume. Weight loss also occurred when the sample was maintained at 395 °C or 450 °C for 20 min. (Supplementary Fig. 4b, c), indicating that the degree of shrinkage could be controlled by modulating the heating temperature while maintaining the heating time. In addition, Fig. 2c, d indicate that SR behaved as a function of the heating temperature and time, which is consistent with the TGA results. The heating temperature used for the volume shrinkage in this work was much lower than that reported in other studies[27,33,36,39]. A lower temperature prevents the transition to glassy carbon with high optical loss, which allows the structures to maintain a relatively low extinction coefficient while increasing the refractive index, which is preferable for photonic applications[35].

Subsequently, we fabricated a woodpile photonic crystal structure consisting of nanoscale feature size to further examine the capabilities of this method. Figure 3 shows SEM images of the as-printed and post-processed woodpile structures on different substrates heated up to and maintained at 395 °C for 5 min. The initial woodpile structure had 13 repeat layers (52 stacks) with initial lattice constants measured as $xy = 1.45\,\mu m$ (nominal $xy = 1.65\,\mu m$), $z = \sqrt{2}xy$, $w = 325\,nm$, and $h = 863\,nm$ (Fig. 3a and Supplementary Fig. 6). A lower laser power and heating temperature were used for the woodpile structure here than for the large 3D models in Fig. 2 to avoid nanoweb formation between the woodpile lines during the printing process[45]. These nanowebs are the narrow and discrete web-like structures that

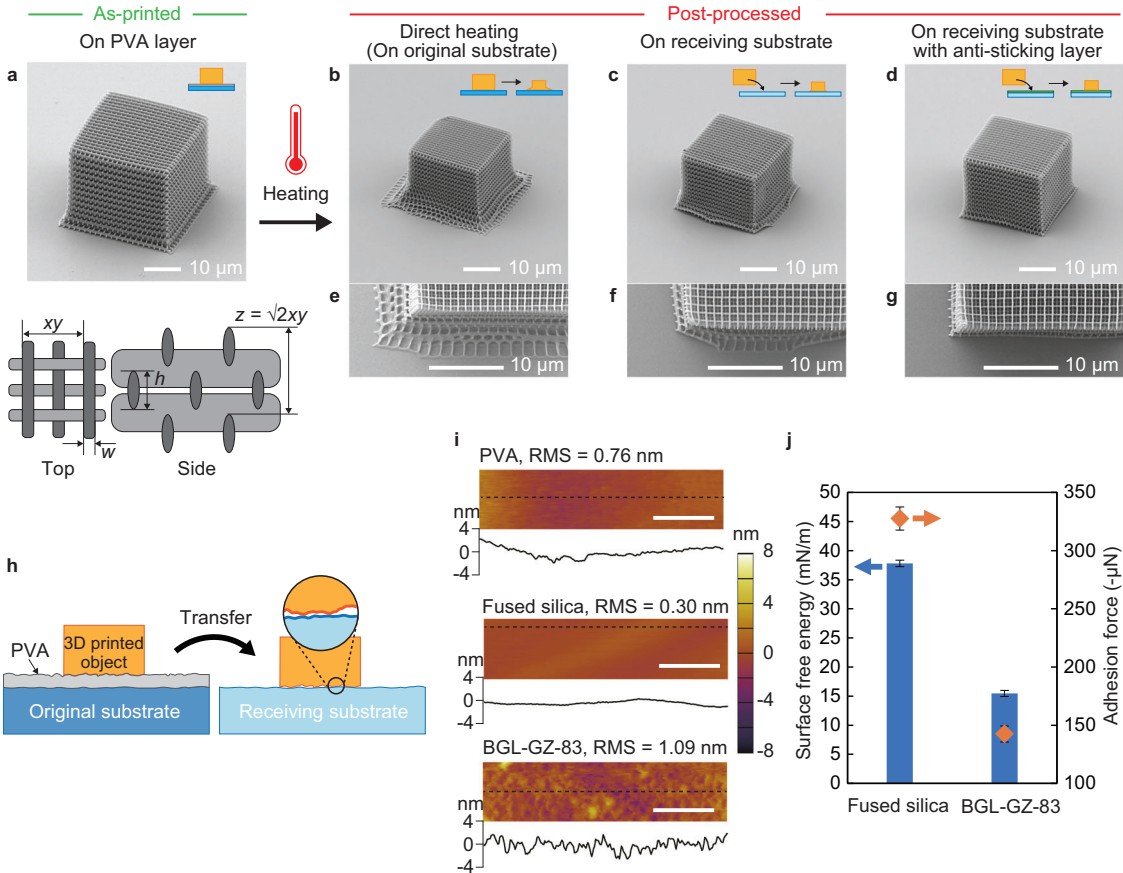

**Fig. 3 | Comparison of shrinkage results on different substrates.** Tilted-view SEM images of **a** as-printed woodpile structure and schematic showing one unit, **b** post-processed woodpile structure using the conventional direct heating method, **c** post-processed woodpile structure on the receiving substrate, and **d** post-processed woodpile structure on the receiving substrate with an anti-sticking layer. **e–g** Top view of each corresponding structure. **h** Schematic of the interface between the transferred 3D model and receiving substrate. **i** AFM images of the PVA layer, bare fused silica substrate, and anti-sticking layer (BGL-GZ-83)-coated fused silica substrate. RMS represents the root mean square. Scale bars = 500 nm. **j** Surface free energy (blue bar) and adhesion force (orange dot) with the receiving substrate for the fused silica substrate and BGL-GZ-83-coated fused silica substrate. The error bars represent standard deviations.

spontaneously form between two closely exposed structures induced by the proximity effect and may greatly affect the optical performance (Supplementary Fig. 7)[20,28,45]. When the two lines are close enough, they may connect with each other or merge into a single unit. The proximity effect will induce laser damage when the excitation light intensity exceeds the energy required for the polymerization of the photoresist. As the laser power increases and the writing speed decreases, the proximity effect becomes more significant; thus, more nanowebs are formed and laser damage is more likely to occur. Hence, in our case, the small woodpile structure was softer and easier to deform than the larger models, as seen in the slight shrinkage of the as-printed wood-pile structure relative to its base layer right after development in Fig. 3a. In the conventional direct heating method (without transfer to the receiving substrate), the bottom part of the structure readily deformed into a net-like structure as the base is strongly attached to the fused silica substrate (Fig. 3b). The deformation of the bottom layers is similar to that in our previous report[35]. Here, the direct heating method produced a lower shrinkage than that in our previous report because we selected a lower heating temperature (395 °C instead of 450 °C) and shorter heating time (5 min instead of 12–21 min) to avoid bond generation between the woodpile lines during heating. Further-more, these samples were heated in air instead of Ar. By contrast, the proposed method demonstrated effective results for uniform shrink-ing on a fused silica substrate as the receiving substrate (Fig. 3c).

The surfaces of the 3D printed structures and substrates generally exhibit nanoscale, microscale, and mesoscale asperities. After the structures were detached from the original substrate and transferred to the receiving substrate, only a small fraction of the surfaces was assumed to have come into contact with each other. This finding indicated that the true contact area was composed of sparse contact points owing to the surface roughness, as shown in Fig. 3[46–50]. The surface roughness and texture of the original and receiving substrates were characterized using an atomic force microscope (AFM) (Fig. 3i). The root mean square (RMS) of the surface roughness of the substrates was 0.3–1.0 nm and showed some asperities, which complemented the schematic in Fig. 3h. Therefore, the van der Waals forces between the transferred 3D structure and receiving substrate can be assumed to have become smaller and easier to overcome than those of the as-printed structure, which adhered to the original substrate with perfect contact. This is also considered using the equation of van der Waals forces per unit area between two surfaces (Supplementary Note 1)[51]. The main mechanism of our shrinkage process is that non-conformal contact between the 3D printed structure and substrate. In the case of the as-printed structure mounted directly on glass, the contact is perfectly conformal. However, once detached, the contact is no longer conformal, which leads to reductions in the adhesion and ability of the printed structures to slide along the substrate surface during shrink-ing. The non-conformal contact between different asperities is closely related to the shrinkage behavior.

Meanwhile, a small distortion in the bottom part was observed when placed on the fused silica as the receiving substrate (Fig. 3c). As the small and fine woodpile structures are more conformal than the

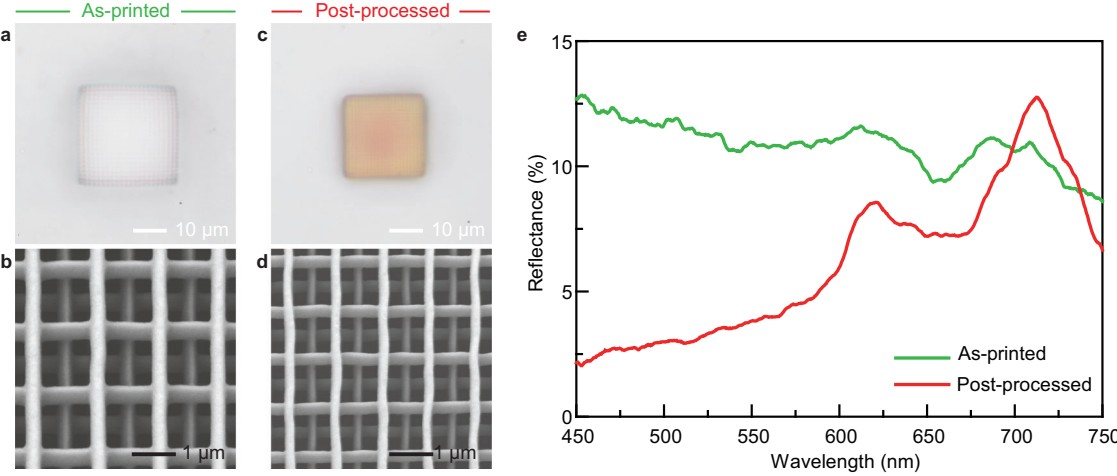

**Fig. 4 | Comparison of the spectral performance before and after post-processing. a, b** Respective top views of bright-field reflective optical and SEM images of as-printed woodpile structures. **c, d** Respective top views of bright-field reflectance optical and SEM images of post-processed woodpile structures. **e** Reflectance spectra of woodpile structures.

larger structures in Fig. 2, regions of the bottom layer inevitably adhered to the substrate during the heating process and caused distortion. Therefore, we changed the receiving substrate to a fused silica substrate treated with an anti-sticking layer to form weaker van der Waals forces between the structures and receiving substrate. We selected BGL-GZ-83 (Profactor GmbH) as the anti-sticking layer because it is the most common material used as a fluorinated surface for nanoimprint lithography molds (Supplementary Note 2)[52]. BGL-GZ-83 is a fluorinated solution that is spin-coated on a substrate and evaporated to form a thin layer of a few nanometers, providing a quick and convenient way to decrease the surface free energy of the substrate without using vapor deposition methods[53,54]. The woodpile structure on the anti-sticking layer successfully achieved uniform shrinkage after heating (Fig. 3d), as affirmed in multiple tests. The results demonstrated that the fabrication reliability and reproducibility of smaller and more sensitive structures were improved using an anti-sticking layer. The top view of the corresponding post-processed woodpiles is presented in Fig. 3e–g to compare the deformed region in detail. The woodpile on the anti-sticking layer clearly exhibited uniform shrinkage while the others did not, and slight deformation of the top structures also appeared owing to the anchoring effect of the bottom substrate (Fig. 3e, f).

The water contact angles of the receiving substrates were measured to further understand the reasons behind the improvement in uniform shrinkage. The receiving substrate with an anti-sticking layer exhibited a larger contact angle (108.1° ± 3.4°) than the fused silica substrate without any coating layer (69.8° ± 3.5°) (Supplementary Fig. 8). In addition, the surface free energy $\gamma_s^{\text{Total}}$ of the receiving substrates was calculated by the following equation by Owens and Wendt:[55]

$$\gamma_s^{\text{Total}} = \gamma_s^{\text{d}} + \gamma_s^{\text{h}} \qquad (2)$$

where $\gamma_s^{\text{d}}$ is the dispersive component of the solid and $\gamma_s^{\text{h}}$ is the hydrogen bond component of the solid. The $\gamma_s^{\text{d}}$ and $\gamma_s^{\text{h}}$ are obtained from the measured contact angles on the receiving substrates and the known surface free energies of two types of liquids (water and diiodomethane). The $\gamma_s^{\text{Total}}$ values of the fused silica substrate and BGL-GZ-83 coated fused silica substrate were 37.8 ± 0.6 and 15.5 ± 0.5 mN/m, respectively. The surface free energy of the BGL-GZ-83 layer was less than half that of the fused silica substrate. Thus, the fluorinated surfaces containing $CF_2$ and $CF_3$ groups present the expected low-surface-energy state[56–58]. Moreover, the physical adhesion force

between two different solid surfaces was measured using a surface-force measurement system (Supplementary Note 3)[59–61]. The receiving substrate with an anti-sticking layer exhibited a lower adhesion force (−142.5 μN ± 9.9 μN) than the pure fused silica substrate (−327.5 μN ± 6.9 μN) (Supplementary Fig. 9). Here, the absolute value represents the strength of the adhesion force because an attractive force is defined as a negative value. We can assume that an anti-sticking layer is relatively easy to separate from a 3D printed structure and a substrate. Using physical and chemical analytical approaches, we found that the interactions between the 3D structure and substrate could lead to the minimization of adhesion force, as shown in Fig. 3j. Hence, we hypothesize that this anti-sticking layer assisted in the uniform shrinking of the bottom of the 3D woodpile by reducing the 3D printed structure–substrate adhesion during the heating process. This observation implies that the base of the 3D objects will easily glide across the anti-sticking layer during shrinkage.

Figure 4 shows SEM images and corresponding optical images under bright-field reflective illumination of the as-printed and post-processed woodpile structures. In the initial state, the as-printed woodpile with lattice distance $xy = 1.45$ μm appeared colorless because its reflectance spectrum was almost flat in the whole visible range, as shown in Fig. 4e. However, the post-processed woodpile structure with $SR = 25$–28% exhibited orange structural colors with two strong resonant peaks (620 and 712 nm) appearing in the spectrum. After post-processing, the lattice constant decreased, whereas the refractive index increased. The main peaks in the visible region can be explained by the presence of different slow-light modes along the Γ−K direction in the first Brillouin zone, resulting in the appearance of reflectance peaks and generating color[35,62]. We have previously shown that stopbands, instead of slow-light modes, would result in more vibrant colors[35]. However, these features can be achieved only with photonic crystals with a smaller lattice constant and different unit cells that result in structures that are too fragile for conventional handling during the transfer process. By contrast, the structures reported in this study have sufficient rigidity; thus, they exhibit less shrinkage. Meanwhile, we found that the post-processed woodpile obtained by conventional direct heating without a transfer process, that is, the structure in Fig. 3b, exhibited poorer color purity and structural stability (Supplementary Fig. 10) than the woodpile obtained with a transfer process. Therefore, the proposed method is advantageous for uniform shrinking.

Finally, to demonstrate the ability of uniform structural color generation in a complex 3D colored macroscale object, we fabricated a

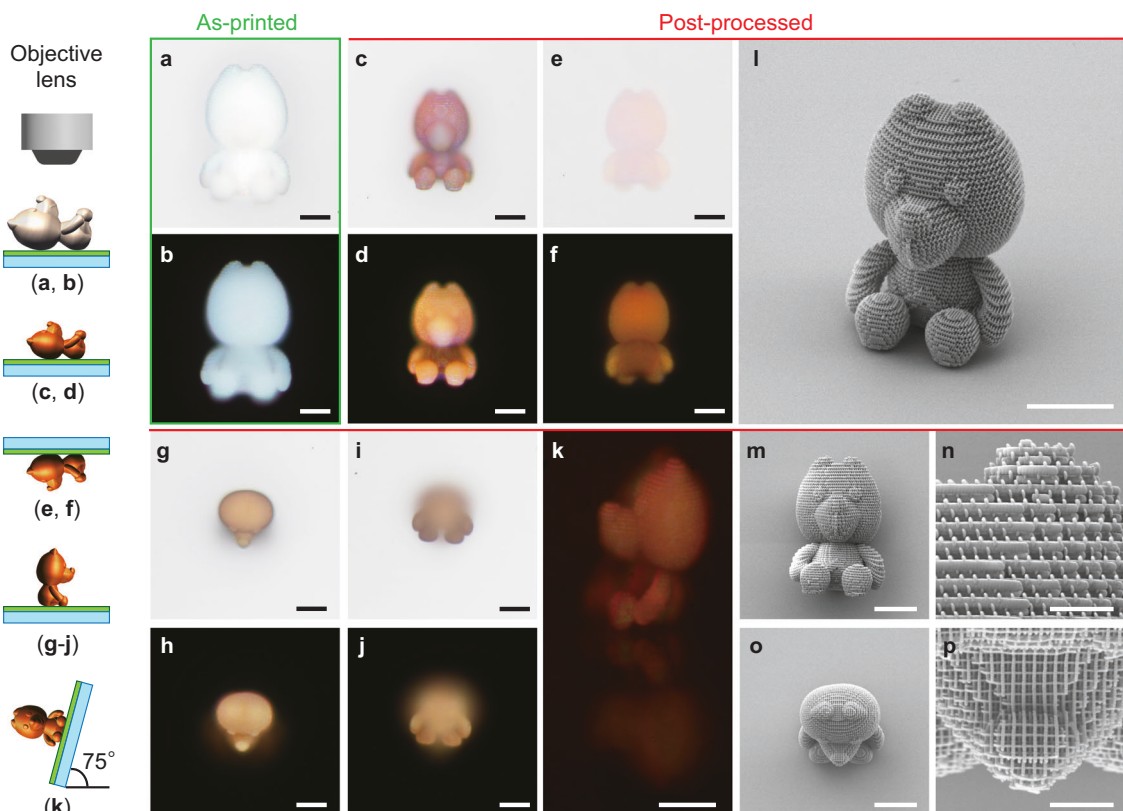

**Fig. 5 | Uniform shrinkage of the 3D photonic crystal mascot with nanoscale features. a** Bright- and **b** dark-field reflective optical images of as-printed 3D mascot. **c, e, g, i, k** Bright- and **d, f, h, j** dark-field reflectance optical images of post-processed 3D mascots viewed from different angles. The 3D mascot lies on the receiving substrate in the front and back views. Back views were obtained by flipping the lying 3D mascot on the receiving substrate. The 3D mascot sat on the receiving substrate in the top view. **l–p** SEM images of post-processed 3D mascots viewed from different angles. Scale bars in **a–m, o** = 20 μm, scale bars in **n, p** = 5 μm.

microscopic 3D model of an iconic mascot of Wakayama Prefecture in Japan called KIICHAN comprising woodpile photonic crystals. The 3D model-shaped photonic crystal structure was generated using an in-house MATLAB code by precisely determining the outer profile of the 3D stereolithographic (STL) model using the ray-tracing method. The internal woodpile structure was determined using a model with pre-designed structural parameters. Subsequently, the file was outputted in the Nanoscribe General Writing Language (GWL) file format with printing parameters suitable for fabrication. The printing parameters of woodpile photonic crystals were set exactly the same as those in Fig. 4. The reflective optical images under bright- and dark-field illuminations in Fig. 5 show the as-printed and post-printed 3D mascots viewed from different angles by preparing two versions (sitting down and lying down). The reflective optical images of the front view of the as-printed 3D mascots showed a light gray color (Fig. 5a, b). By contrast, the post-processed 3D mascots had accurate and uniform shapes without sacrificial support structures, remained intact after thermal shrinkage ($SR = 28\%$, shrinkage of the major axis to <70 μm), and exhibited vivid colors. The structural color of the front view of the 3D mascot was slightly reddish, unlike that of the top view, because the lattice pattern was different, and the lattice distance of the front view was larger than that of the top view, as shown in Fig. 5m–p and Supplementary Fig. 6. Moreover, specific photonic cavity modes were excited under reflective illumination in the front view. These modes are produced by coupling the cavity oscillation and high-order propagation modes along the woodpile lines owing to the different designs compared with the top view[62]. The back view images in Fig. 5e, f were obtained by flipping the receiving substrate with the lying 3D mascot on top, meaning that Fig. 5e, f were imaged from the opposite side passing through the substrate. The 3D model did not fall off the

substrate, indicating sufficient van der Waals forces attaching it to the receiving substrate. However, it was a negligibly small contact area with no influence on uniform shrinking because no deformation was observed (Supplementary Fig. 11). The structural color under the dark-field did not differ significantly at different angles because of the ring-shape illumination and detection of scattered light from the 3D mascot (Fig. 5d, f, h, j). Figure 5k shows that the high-magnification side view of the bright-field reflective optical image and structural color exhibits significant spatial uniformity, which is further cross validated from the SEM images of uniform woodpile photonic crystal structures (Fig. 5l–p and Supplementary Fig. 12).

## Discussion

The developed PVA assisted pick and place process can be applied to easily overcome the resolution limit of 3D printed structures and obtain uniform optical properties, as demonstrated by the vivid structural colors of the printed woodpile photonic crystal structures. We demonstrated uniform shrinkage of 3D printed structures of different sizes and shapes using the unique properties of PVA. In particular, an anti-sticking layer on the receiving substrate promotes shrinkage uniformity, which is crucial only for structures with submicron features. A complex 3D colored mascot composed of woodpile photonic crystals was successfully fabricated and shrunk to 70 μm in size (major axis). This process is likely extendable to obtain full colors and arbitrary complex designs for use in optics and photonics applications and other research fields requiring nanoscale uniform 3D structures. In future research, the design of a simplified system should be investigated. For example, the thermal shrinkage of 3D micro-objects may be preemptively performed in high-heat-resistance solution materials, such as phosphonium ionic liquids, because the final

step can be eliminated using the receiving substrate. Investigation of a uniform shrinking process using direct heating without a transfer process is also a worthwhile endeavor.

## Methods

### Materials

The IP-Dip2 photoresist was purchased from Nanoscribe GmbH. PVA (Mw31000–50000, 98–99 mol% hydrolyzed), propylene glycol monomethyl ether acetate (PGMEA, 99.5%), isopropyl alcohol (IPA, 99.5%), and nonafluorobutyl methyl ether (NME, 99%) were purchased from Sigma-Aldrich. Hydrophilic polytetrafluoroethylene (PTFE) membrane filters with 0.45 μm pore size were purchased from JVLAB. Fused silica substrates (25 mm squares with a thickness of 0.7 mm) were purchased from Haian Huihong Photoelectric Instrument Factory. The anti-sticking material (BGL-GZ-83) was purchased from Profactor GmbH.

### Fabrication method

A 4 wt.% PVA aqueous solution was spin-coated onto a fused silica substrate at 4000 rpm for 20 s with 2000 rpm/s acceleration ramp and then baked at 80 °C for 2 min. The thickness of the film was 90–100 nm. Subsequently, 3D objects were printed on top of the PVA layer using a TPL system (Nanoscribe GmbH, Photonic Professional GT2) and IP-Dip2 photoresist was used as a negative photoresist for the dip-in laser lithography configuration. Pattern files for 3D printing determined the confined positions of the focal point of laser for polymerization. A 780 nm femtosecond pulsed laser was used and controlled to a laser power of 27.5–30 mW at a speed of 40 mm/s for the 3D models (Fig. 2) and a laser power of 23–25.5 mW at a speed of 9 mm/s for the small woodpile structures and 3D mascots (Figs. 3, 4, 5). The unexposed photoresist was removed by immersion in PGMEA for 20 min, IPA for 3 min, and NME for 5 min. UV irradiation (MX-150, Dymax) at 405 nm with 75% maximum power was performed during immersion in IPA. After development, the sacrificial PVA layer was dissolved in water. The 3D printed objects were captured using a hydrophilic PTFE membrane filter. They were then transferred to a receiving substrate and heated up in air at a rate of 10 °C/min and maintained at 450 °C for 5 min for the 3D models (Fig. 2) and 395 °C for 5 min for the small woodpile structures and 3D mascots (Figs. 3, 4, 5). A fused silica substrate and an anti-sticking layer-coated fused silica substrate were used as the receiving substrates. BGL-GZ-83, as the anti-sticking layer, was spin-coated on a fused silica substrate at 1000 rpm for 30 s, followed by 2000 rpm for 30 s with a 1600 rpm/s acceleration ramp, and then used after 8 h.

### Reflectance measurement

Bright-field and dark-field reflectance optical images and reflectance spectra were acquired using an optical microscope (Eclipse LV100ND, Nikon) equipped with a color CMOS camera (DS-Ri2, Nikon) and microspectrophotometer (508 PV, CRAIC). The 3D structures were illuminated using a halogen lamp and were captured in reflection mode with an objective lens (×20/0.45 NA for the 3D models and 3D mascot, ×50/0.4 NA long working distance for the small woodpile structures and tilted view angle imaging of the 3D mascot). The reflectance spectra of the 3D structures were measured in reflection mode with a ×20/0.45 NA objective lens and were normalized to that of a silver mirror (PF20-03-P01, Thorlabs). The measurements were performed under the same conditions used for the 3D structures.

### Characterization

SEM images were obtained using a JSM-7600F (JEOL) instrument at an accelerating voltage of 5 kV in lower secondary electron detection mode. The water contact angles and surface free energies were obtained using a contact angle meter (DMo-502, Kyowa Interface Science). The surface free energies of the receiving substrates were calculated using the contact angles of water and diiodomethane, and the

theoretical equation by Owens and Wendt[55]. TGA was performed using a TGA Q50 instrument (TA Instruments) with a Pt pan to hold the sample. The analysis was performed in a closed chamber with an air flow of 40 ml/min at a temperature increase rate of 10 °C/min. The surface roughness and texture were measured using an AFM (MFP-3D Origin, Asylum Research) at a scan rate of 0.2 Hz. The adhesion forces of the receiving substrates were measured using an ENT-5X instrument (ELIONIX Inc.) at a stage speed of 200 nm/s and load step of 1 μN (20 ms intervals).

## Data availability

All data generated in this study are provided in the published article and the corresponding supplementary information files. All data and source data for Figs. 2c, d and 3j are available from the corresponding authors upon request.

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

## Acknowledgements

J.K.W.Y. would like to acknowledge funding support from the National Research Foundation (NRF), Singapore, under the Competitive Research Programme (Grant No. NRF-CRP20-2017-0004) and NRF Investigatorship (Grant No. NRF-NRFI06-2020-0005). Tomohiro M., Takeshi M., and H.K. are grateful for the special support of the Research for Advanced

Core Technologies (ReACT) in Wakayama Prefecture, Japan. The authors would like to thank ELIONIX Inc. for measuring the adhesion forces of the receiving substrates.

## Author contributions

Tomohiro M., H.W., and J.K.W.Y. contributed to the conceptualization and methodology of the experiments. Tomohiro M., H.W., and C.C.S. designed and performed experiments. D.A., J.N., M.A.R., and Takeshi M. provided technical guidance and validated the results. W.Z. and H.L. conducted the thermogravimetric analysis, and Tomohiro M. and Takeshi M. conducted the surface analysis. Tomohiro M., H.W., W.Z., C.-F.P., and H.K. designed and printed the 3D models. All authors contributed to the validation of all data and writing (review and editing) of the manuscript.

## Competing interests

The authors declare no competing interests.
