## [Peer Review File · Nature Communications]

Pick and Place Process for Uniform Shrinking of 3D Printed Micro- and Nano-Architected MaterialsREVIEWER COMMENTS

Reviewer #1 (Remarks to the Author):

In this manuscript, 3D printed structures with uniform heat-shrinking are achieved by exploiting poly(vinyl alcohol) (PVA)-assisted pick-and-place process, which solves the issues of limited resolution and undesirable base distortion caused by the substrate pinning in the conventional direct heating process. The water-soluble PVA layer spin-coated on the original substrate facilitates the structure to detach and transfer to the receiving substrate. Through regulating the surface roughness and free energy of the receiving substrate, the adhesion force can be minimized and promote shrinkage uniformity, thus realizing the fabrication of uniform woodpile photonic crystal structure with nanoscale features. The implementation of the proposed approach and the demonstrations are interesting, while some statements in the manuscript are not well supported. Therefore, a revision is required before it could be published in Nature Communications.

1. Please provide corresponding data, for example, the optical, SEM images and the dimensional change ratio before and after being heated from the different testing directions, to clearly demonstrate the statement in Line 141, Page 7 “with the same degree of shrinkage along each axis of the structure (Figure 2(a) and (b))”. In addition, the uniform mechanical property obtained after pyrolytic shrinkage has been mentioned, but it still lacks corresponding data to support the conclusion.
2. In the thermogravimetric experiment, what is decomposed during the large weight loss stage occurred at 350–450 °C?
3. The mechanism of the uniform shrinkage is due to the smaller van der Waals forces between the contact surfaces of transferred 3D structure and receiving substrate. It can be seen from Figure 3(h), the contact surfaces of the 3D structure are influenced by the surface roughness of the PVA layer. Thus, besides the properties of the receiving substrate, whether the roughness of the PVA also determine the shrinkage results?
4. The post-processed woodpile structure suffers from uniform shrinkage, why does the reflectance spectrum have two peaks (Figure 4(e))? The optical image in Figure 4(c) also exhibits nonuniform structural color, with red in center and orange in periphery.
5. Some details need to be verified and revised.
(1) The description in Line 204, Page 10 is “However, a small distortion in the bottom part

was observed when placed on the fused silica as the receiving substrate (Figure 3 (b)).”

Please verify the figure label, it may be Figure 3(c), not Figure 3 (b).

(2) Abbreviations need to be spelled out for the first time, for example the PTFE in Line 343, Page 17.

(3) In the “Fabrication method” section, what is the purpose of immersing the as-printed sample in IPA along with the UV irradiation? Whether the UV irradiation will initiate the unintended curing and reduce precision?

Reviewer #2 (Remarks to the Author):

First of all, thank you for the opportunity to review the manuscript written by Joel Yang et al.

I personally enjoyed the manuscript.

This manuscript is a follow-up to a paper previously published by the authors in Nature Communications (Ref #35).

The purpose of this study is to technically solve the non-uniform shrinkage in the vertical direction that occurs when thermally shrinking 3D micro-objects printed through a well-known commercial TPL machine (Nanoscribe). I believe that researchers in this field can agree on the importance of this technical issue. To address this issue, the authors proposed a transfer process that utilizes a PVA sacrificial layer to weaken the physical interaction between the substrate and the 3D printed micro-objects. To strengthen the theoretical aspect, an analysis based on the surface roughness of the receiving substrate was partially performed. A 3D mascot model with uniform structural color was successfully demonstrated.

Overall, the manuscript was faithfully performed in terms of experiments and was technically well described.

The only concern is whether this manuscript conforms to Nature Communications' standards. The scientific nitty-gritty of scaling down 3D printed micro-objects via heat shrinking has already been covered with a nice demonstration in the author's previous paper. Most of the scientific points related to the scale-down of 3D printed micro-objects

via heat shrink have already been covered in the author's previous paper with nice demonstrations. Although this study addresses an important issue related to anisotropic shrinkage within their process, microneedle-based transfer is not promising or makes follow-up studies difficult. This makes the paper suitable for more specific journals. Anyway, I'd like to leave this to the editor's discretion.

Here are some of the issues that need to be addressed in terms of content.

1. Can't thermal shrinking of 3D micro-objects be preemptively performed in an intermediate step? For example, isotropic shrinkage of 3D micro-objects is likely possible on a membrane or in a solution bath, but not necessarily on a receiving substrate. Of course, it will be necessary to select the optimal membrane or solution material considering the high-temperature process around 400 degrees. Is it possible to heat-shrink large quantities of 3D micro-objects first and then transfer them one by one to the desired substrate?

2. In-depth analysis of the physical adhesion between the receiving substrate and 3D micro-objects is lacking. More quantitative analysis and discussion are needed. For example, it is necessary to derive a critical van der Waals force capable of preventing detachment from a receiving substrate without causing non-uniform shrinkage of the 3D micro-object in the vertical direction upon thermal shrinkage.

3. It is necessary to experimentally measure the adhesion force between the 3D micro-object and the receiving substrate before and after thermal shrinkage.

4. In Figure 4, the occurrence of the photonic bandgap at visible wavelengths after heat treatment can be easily predicted through the author's previous paper. Rather, it is necessary to present the difference in reflection behavior of 3D photonic crystals depending on whether or not there is a transfer process. That is, the advantage of uniform shrinkage in the vertical direction should be intensively discussed from an optical point of view.

5. One-by-one transfer of 3D micro-objects to a receiving substrate using microneedles may be an interesting demonstration, but it is very inefficient and lacks prospects in terms of

application. In particular, since TPL is one of the fundamentally low-productivity processes, the follow-up process proposed in this study further aggravates this distrust. Can the authors speed up the transfer process or suggest suitable applications for their process?

Reviewer #3 (Remarks to the Author):

The authors present in this manuscript a new method based on using poly(vinyl alcohol) (PVA) as a sacrificial layer between the substrate and the printed structure, to avoid the deformation caused by the strong attachment between the original substrate and the prints. Subsequently, the 3D structures on the receiving substrate were heated and shrunk without distortion to obtain a uniformly scaled version of the original shape.

The manuscript is very well written, and suitable for publication in Nature Communications after addressing the minor comments:

- The process includes partial thermal decomposition , at high temperature under air. About 80% of the material decomposes at about 450C, which is not very common to most organic materials. Therefore, the chemical composition of the photoresist should be given.
- Page 8, figure 2: this is the first time laser power was mentioned. The contact and importance of these values should be explained in the manuscript before the figure capture.
- Page 9, lines 174-177 "Lower laser power and heating temperature were used for the woodpile structure here than for the large 3D models in Figure 2 to avoid nanoweb formation between the woodpile lines during the printing process. These nanowebbs are induced by the proximity effect and may greatly affect the optical performance (Figure S4)." Clear definitions are required for "nanoweb" and "proximity effect" as they are not adequately defined here and in Figure S4. The lack of clarity leads to a lack of understanding of the described.
- Page 9, line 194, "only a tiny fraction of the surfaces came in contact with each other" Calculating the (normalized) contact area is recommended instead of relying on the articular description. This will provide a more accurate representation of the differences.

Reviewer Comments and Responses

Reviewer 1:

General Comment:

In this manuscript, 3D printed structures with uniform heat-shrinking are achieved by exploiting poly(vinyl alcohol) (PVA)-assisted pick-and-place process, which solves the issues of limited resolution and undesirable base distortion caused by the substrate pinning in the conventional direct heating process. The water-soluble PVA layer spin-coated on the original substrate facilitates the structure to detach and transfer to the receiving substrate. Through regulating the surface roughness and free energy of the receiving substrate, the adhesion force can be minimized and promote shrinkage uniformity, thus realizing the fabrication of uniform woodpile photonic crystal structure with nanoscale features. The implementation of the proposed approach and the demonstrations are interesting, while some statements in the manuscript are not well supported. Therefore, a revision is required before it could be published in Nature Communications.

Response:

We thank the reviewer for their positive comments and for finding our proposed approach and demonstrations interesting. Revisions and additional data are provided to support our findings, as detailed below.

Comment 1:

Please provide corresponding data, for example, the optical, SEM images and the dimensional change ratio before and after being heated from the different testing directions, to clearly demonstrate the statement in Line 141, Page 7 “with the same degree of shrinkage along each axis of the structure (Figure 2(a) and (b))”. In addition, the uniform mechanical property obtained after pyrolytic shrinkage has been mentioned, but it still lacks corresponding data to support the conclusion.

Response 1:

We appreciate the reviewer pointing out that our claim of uniform shrinkage was not strongly supported by the data provided in the previous version of our manuscript. Uniform shrinkage can be simply observed from the optical and SEM images in Figure 2. To further address the reviewer’s concern and quantify the shrinkage rate (SR) in 3D, we measured the dimensional change ratio as SR along all directions and determined the

lattice constant using SEM images of a simple cubic model. We clearly observed the same degree of shrinkage along each axis. We have included this additional data in the Supplementary Information (Figure S2).

Figure S2. Measurement of the shrinkage rate (SR) in different directions. The 3D simple cubic model was heated up to and maintained at 450 °C for 5 min on the receiving substrate. (a) SEM images of the as-printed and post-processed simple 3D cubic models. The bottom images show side views of the models tilted by 55°. (b) SR in each direction and the corresponding lattice constant. (c) Schematic of the calculation of the actual length of S_{a-z} and S_{p-z} (S_z in the SEM image/ $\sin 55^\circ$).

Regarding the reviewer’s comments on mechanical properties, other groups have reported the mechanical characterization of uniform shrinkage structures with sacrificial support structures (*Nat. Mater.* **15**, 438–443 (2016) and *Nat. Commun.* **9**, 593 (2018)). The uniform nondistorted structures that these groups used for their compression experiments allowed for an accurate evaluation of mechanical properties because the compressive stress is applied to the structures uniformly. We believe that the same effect is present in our approach, which has the critical advantage of not requiring support structures. In addition, because we are focusing on the optical properties of the structures in our study, we have de-emphasized their mechanical aspects in the manuscript. A thorough investigation of uniform mechanical properties could be the focus of a separate work. Accordingly, we have removed the term “mechanical” from the Conclusion section to avoid confusion (page 19, line 363).

“The developed PVA-assisted pick-and-place process can be applied to easily overcome the resolution limit of 3D-printed structures and obtain uniform optical properties, as demonstrated by the vivid structural colors of the printed woodpile photonic crystal structures.”

Comment 2:

In the thermogravimetric experiment, what is decomposed during the large weight loss stage occurred at 350–450 °C?

Response 2:

According to the vendor-supplied datasheet, the IP-Dip2 photoresist that we used consists of 60–80% 2-(hydroxymethyl)-2-[[[(1-oxoallyl)oxy]methyl]-1,3-propanediyl diacrylate (CAS No. 3524-68-3, common name: pentaerythritol triacrylate). The large weight loss observed at 350–450 °C is associated with the onset of carbonization. During the heating process, the formation of volatile species such as CO and CO₂ by carbonization causes a large volume loss. We also analyzed the Raman spectra of the photoresist before and after heating. Thermolysis reduced the intensities of the peaks associated with the C–H (2947 cm⁻¹), C=O (1722 cm⁻¹), and C–O (935 cm⁻¹) stretching modes of the photoresist. In contrast, peaks corresponding to sp²-rich carbon (2500–3100 cm⁻¹), graphitic carbon (1593 cm⁻¹), and disordered carbon (1353 cm⁻¹) were observed. These results indicate that the photoresist formed an activated carbon-like structure with increased carbon content.

We have added the relevant explanation and additional references to the manuscript (page 7, line 149–page 8, line 160; text highlighted in yellow).

“A large weight loss was observed at 350–450 °C, which was associated with carbonization, followed by a smaller peak at 540 °C, which was associated with the decomposition of polymeric chains.⁴⁰⁻⁴² The IP-Dip2 photoresist is composed mainly of pentaerythritol triacrylate, which contains a large fraction of bonded oxygen. During the heating process, the formation of volatile species such as CO and CO₂ by carbonization causes a large volume loss. We also analyzed the Raman spectra of the photoresist before and after heating (Figure S4). Thermolysis reduced the intensities of the peaks associated with the C–H (2947 cm⁻¹), C=O (1722 cm⁻¹), and C–O (935 cm⁻¹) stretching modes. In contrast, peaks corresponding to sp²-rich carbon (2500–3100 cm⁻¹), graphitic carbon (1593 cm⁻¹), and disordered carbon (1353 cm⁻¹) were observed.^{43, 44} These results

indicate that the photoresist formed an activated carbon-like structure with increased carbon content.”

We have also added the Raman data to the Supplementary Information (Figure S4).

Figure S4. Raman spectrum of the IP-Dip2 photoresist, which consists of 60–80% 2-(hydroxymethyl)-2-[[1-(1-oxoallyl)oxy]methyl]-1,3-propanediyl diacrylate (CAS No. 3524-68-3, common name: pentaerythritol triacrylate).

Comment 3:

The mechanism of the uniform shrinkage is due to the smaller van der Waals forces between the contact surfaces of transferred 3D structure and receiving substrate. It can be seen from Figure 3(h), the contact surfaces of the 3D structure are influenced by the surface roughness of the PVA layer. Thus, besides the properties of the receiving substrate, whether the roughness of the PVA also determine the shrinkage results?

Response 3:

As the reviewer points out, we assumed that the surface roughness of the PVA layer influenced the shrinkage result, as the roughness of the bottom side of the 3D-printed structure transferred the surface shape of the PVA layers. However, this effect is expected to be relatively small. When we performed shrinkage to remount different faces of the 3D-printed structures on the receiving substrate, we observed no significant difference in the shrinkage rates. Therefore, the transfer process, that is, simply placing the object on the receiving substrate with low interaction forces between them, and the surface properties of the receiving substrate have a more profound effect on the shrinkage results, as demonstrated in Figures 3 and S7.

Comment 4:

The post-processed woodpile structure suffers from uniform shrinkage, why does the

reflectance spectrum have two peaks (Figure 4(e))? The optical image in Figure 4(c) also exhibits nonuniform structural color, with red in center and orange in periphery.

Response 4:

These two peaks can be explained by the presence of slow-light modes in the photonic band structure along the Γ -K direction in the first Brillouin zone, as the lattice constant was larger than that in a previous report (*Nat. Commun.* **10**, 4340 (2019)). We have previously shown that stopbands, instead of slow-light modes, would result in more vibrant colors. However, these features can be achieved only with photonic crystals with a smaller lattice constant and different unit cells that result in structures that are too fragile for conventional handling during the transfer process. By contrast, the structures reported in our study have sufficient rigidity and, thus, exhibit less shrinkage. The small color shift of the structure may be due to the edge effect and nanoweb formation between the printed lines. Further improvements in the printing conditions and transfer process to circumvent these shortcomings could be a topic for future studies. We have added the above explanation to the Results and Discussion section (page 15, lines 294–302, text highlighted in yellow).

“The main peaks in the visible region can be explained by the presence of different slow-light modes along the Γ -K direction in the first Brillouin zone, resulting in the appearance of reflectance peaks and generating color^{35,61}. We have previously shown that stopbands, instead of slow-light modes, would result in more vibrant colors³⁵. However, these features can be achieved only with photonic crystals with a smaller lattice constant and different unit cells that result in structures that are too fragile for conventional handling during the transfer process. By contrast, the structures reported in this study have sufficient rigidity; thus, they exhibit less shrinkage.”

Comment 5:

Some details need to be verified and revised.

(1) The description in Line 204, Page 10 is “However, a small distortion in the bottom part was observed when placed on the fused silica as the receiving substrate (Figure 3 (b)).” Please verify the figure label, it may be Figure 3(c), not Figure 3 (b).

(2) Abbreviations need to be spelled out for the first time, for example the PTFE in Line 343, Page 17.

(3) In the “Fabrication method” section, what is the purpose of immersing the as-printed sample in IPA along with the UV irradiation? Whether the UV irradiation will initiate the unintended curing and reduce precision?

Response 5:

Thank you for your careful review and comments. We have incorporated the following changes:

(1) Before: Figure 3(b), Corrected: Figure 3 (c) (page 11, line 226; text highlighted in yellow)

(2) Before: PTFE, Corrected: polytetrafluoroethylene (PTFE) (page 19, line 384; text highlighted in yellow)

(3) PGMEA, which is used in the first step of the conventional development process, removes unbound monomeric materials. In the second step, the 3D structures are rinsed with IPA. In our work, we adopted additional UV irradiation in this step. This curing process enables further cross-linking between the reactive sites of the photoresist molecules, which remained unreacted after the initial laser exposition inside the structure. The stability of the structures is improved by UV irradiation. Some papers have reported this procedure; for example, Purto et al. investigated and reported this effect in detail (*Microelectron. Eng.* **194**, 45–50 (2016)).

Reviewer 2:

General Comment:

First of all, thank you for the opportunity to review the manuscript written by Joel Yang et al. I personally enjoyed the manuscript. This manuscript is a follow-up to a paper previously published by the authors in Nature Communications (Ref #35). The purpose of this study is to technically solve the non-uniform shrinkage in the vertical direction that occurs when thermally shrinking 3D micro-objects printed through a well-known commercial TPL machine (Nanoscribe). I believe that researchers in this field can agree on the importance of this technical issue. To address this issue, the authors proposed a transfer process that utilizes a PVA sacrificial layer to weaken the physical interaction between the substrate and the 3D printed micro-objects. To strengthen the theoretical aspect, an analysis based on the surface roughness of the receiving substrate was partially performed. A 3D mascot model with uniform structural color was successfully demonstrated. Overall, the manuscript was faithfully performed in terms of experiments and was technically well described. The only concern is whether this manuscript conforms to Nature Communications' standards. The scientific nitty-gritty of scaling down 3D printed micro-objects via heat shrinking has already been covered with a nice demonstration in the author's previous paper. Most of the scientific points related to the scale-down of 3D printed micro-objects via heat shrink have already been covered in the author's previous paper with nice demonstrations. Although this study addresses an important issue related to anisotropic shrinkage within their process, microneedle-based transfer is not promising or makes follow-up studies difficult. This makes the paper suitable for more specific journals. Anyway, I'd like to leave this to the editor's discretion.

Response:

We are thankful that the reviewer enjoyed reading our manuscript and found it “faithfully performed in terms of experiments and was technically well described.” As the reviewer points out, the current TPL technology presents several limitations in terms of, for example, material types, structural feature sizes, and mechanical properties. Scaling down 3D-printed micro-objects via heat shrinking was previously proposed to tackle these issues, leading to changes in the material composites, reductions in feature size, and enhanced structural properties. This method has been well investigated and is employed in various research fields. However, as summarized in our Introduction, most structures suffer from nonuniform shrinkage during heating, and the addition of sacrificial supporting structures introduces new problems that must also be resolved. Hence, as the reviewer comments, our study addresses an important issue related to anisotropic shrinkage during the heating process. Many studies will benefit from the micro/nanoscale

uniform structures obtained via our method, as they can be used in applications such as visible/near-infrared 3D topological optics, quantum optics, optical chips, metasurfaces, and so on. Beyond micro and nano optics, our structures may also bring about new possibilities for micro/nanoscale mechanics, biomedical robots, micro/nanofluidics, and thermal management devices, among others. Although microneedle-based transfer was used in our study owing to the small size of the structures, the proposed pick-and-place method provides a promising solution for shape-preserving transfer and heat-based uniform shrinkage, which could be coupled with other post-processing fabrication procedures. In addition, an automatically controlled pick-up system with precise microneedle movement could provide easier transfer. We hope that, inspired by this method, new noncontact transfer methods will be developed in the future, thereby rendering the process simpler and more efficient.

Comment 1:

Can't thermal shrinking of 3D micro-objects be preemptively performed in an intermediate step? For example, isotropic shrinkage of 3D micro-objects is likely possible on a membrane or in a solution bath, but not necessarily on a receiving substrate. Of course, it will be necessary to select the optimal membrane or solution material considering the high-temperature process around 400 degrees. Is it possible to heat-shrink large quantities of 3D micro-objects first and then transfer them one by one to the desired substrate?

Response 1:

We thank the reviewer for their questions and important advice. We indeed attempted to heat-shrink the membrane filter; however, the membrane shrank and became deformed. As the reviewer mentions, the high heat resistance of solution materials must be considered for a simplified system design. Ionic liquids may be used in solution baths. For example, tributyl phosphine with a bis(trifluoromethyl sulfonyl) amide anion was reported to have a high thermal decomposition temperature of over 400 °C (*Electrochem. Commun.* **13**, 178–181 (2011)). The use of water-soluble materials that have a lower decomposition temperature than the photoresist is also possible. If a suitable layer under 3D-printed structures begins to decompose at a lower temperature than the actual structure (i.e., the cured photoresist), the interaction between the objects and substrate may be reduced prior to the heat shrinking of the structure. Therefore, although direct heating could be used to eliminate the transfer process, the effect of such treatment on uniform shrinkage requires further careful study.

To address these possibilities, we have added potential directions for future research to the Conclusion section (page 19, lines 372–377; text highlighted in yellow).

“In addition, the design of a simplified system should be investigated. For example, the thermal shrinkage of 3D micro-objects is preemptively performed in high-heat-resistance solution materials, such as phosphonium ionic liquids, because the final step can be eliminated using the receiving substrate. Investigation of a uniform shrinking process using direct heating without a transfer process is also a worthwhile endeavor.”

Comment 2:

In-depth analysis of the physical adhesion between the receiving substrate and 3D micro-objects is lacking. More quantitative analysis and discussion are needed. For example, it is necessary to derive a critical van der Waals force capable of preventing detachment from a receiving substrate without causing non-uniform shrinkage of the 3D micro-object in the vertical direction upon thermal shrinkage.

Comment 3:

It is necessary to experimentally measure the adhesion force between the 3D micro-object and the receiving substrate before and after thermal shrinkage.

Responses 2 and 3:

We thank the reviewer for their comments. To address the reviewer’s concerns, we performed additional measurements of the physical adhesion force using a surface-force measurement system (ENT-5X, ELIONIX Inc). This quantitative analysis can determine the displacement and pull-off force between the measurement probe (dimethylpolysiloxane-coated spherical glass probe) and the samples (substrates in our case) upon their contact and separation. The pull-off force is the force that exceeds the adhesion force between the probe and sample surface. This measurement reflects the adhesion force between two different solid surfaces with ultra-high resolution (0.03 nN and 0.3 pm), but it was performed at room temperature; thus, it does not indicate the forces during heating. The receiving substrate with an anti-sticking layer exhibited a lower adhesion force ($-142.5 \mu\text{N} \pm 9.9 \mu\text{N}$) than the fused silica substrate without any coating layer ($-327.5 \mu\text{N} \pm 6.9 \mu\text{N}$). Here, the absolute value represents the strength of the adhesion force because an attractive force is defined as a negative value. Because the adhesion force of the anti-sticking layer-coated fused silica substrate is lower than that of the pure fused silica substrate, the two solid-state materials are relatively easier to separate. This finding agrees well with the water contact angle and surface free energy measurements. In addition, the adhesion force can be used as an index of the uniform

shrinkage of the 3D-printed object on the receiving substrate; for example, if the adhesion force between the measurement probe and receiving substrate is under 200 μN (absolute value), the 3D object can be shrunk uniformly.

By comparison, because it cannot be moved by the microneedle, the as-printed 3D structure (without transfer) appears to make contact with the substrate with a high adhesion force owing to the formation of some bond. If the structures are intentionally moved by the microneedle, they break and bounce off. However, we can easily move the post-processed 3D structure on the receiving substrate after heating.

To address this issue, we have added an explanation and new data to the manuscript (page 13, lines 261–270; text highlighted in yellow). We have also modified the data in Figure 3 (j) to show the relationship between surface free energy and adhesion force instead of water contact angles.

“Moreover, the physical adhesion force between two different solid surfaces was measured using a surface-force measurement system (Supplementary Note 3).⁵⁸⁻⁶⁰. The receiving substrate with an anti-sticking layer exhibited a lower adhesion force ($-142.5 \mu\text{N} \pm 9.9 \mu\text{N}$) than the pure fused silica substrate ($-327.5 \mu\text{N} \pm 6.9 \mu\text{N}$) (Figure S8). Here, the absolute value represents the strength of the adhesion force because an attractive force is defined as a negative value. We can assume that an anti-sticking layer is relatively easy to separate from a 3D-printed structure and a substrate. Using physical and chemical analytical approaches, we found that the interactions between the 3D structure and substrate could lead to the minimization of adhesion force, as shown in Figure 3(j).

(j) Surface free energy (blue bar) and adhesion force (orange dot) with the receiving substrate for the fused silica substrate and BGL-GZ-83-coated fused silica substrate.”

Finally, we have added further explanations, references, and raw data of the surface-force measurements to the Supplementary Information (Supplementary Note 3 and Figure S8).

“Supplementary Note 3

Measurement of the adhesion force⁷⁻⁹

The adhesion force was measured using a surface-force measurement system (ENT-5X, ELIONIX Inc.). The surface force can be considered to represent the strength of the physical adhesion between two solid materials, unlike the water contact angle measurements employed in the chemical approach⁷⁻⁹. As illustrated in Figure S7 (a), the measurement procedure is as follows. (i) The measurement probe (dimethylpolysiloxane-coated spherical glass probe) with a spring and sample (substrates in our case) are brought near each other. (ii) When the attractive force between the probe and sample exceeds the spring force of the probe, the probe is attracted by the sample, and the two surfaces come into contact. (iii) The probe is pulled off from the sample using an electromagnetic force. (iv) The displacement and pull-off force of the probe are measured with ultra-high resolution (0.03 nN and 0.3 pm). The pull-off force is the force that exceeds the adhesion force between the probe and sample surface. The receiving substrate with an anti-sticking layer exhibited a lower adhesion force ($-142.5 \mu\text{N} \pm 9.9 \mu\text{N}$) than the fused silica substrate without any coating layer ($-327.5 \mu\text{N} \pm 6.9 \mu\text{N}$) (Figure S7 (b)). Here, the absolute value represents the strength of the adhesion force because an attractive force is defined as a negative value. Because the adhesion force of the anti-sticking layer (BGL-GZ-83)-coated fused silica substrate is lower than that of the pure fused silica substrate, the two solid-state materials are relatively easy to separate. These findings support the water contact angle and surface free energy measurement results.”

Figure S8. (a) Schematic of the adhesion force measurement. (b) Adhesion forces of the anti-sticking layer (BGL-GZ-83)-coated and pure fused silica substrates. The largest absolute value of the load before the displacement jump was determined as the adhesion force.

Comment 4:

In Figure 4, the occurrence of the photonic bandgap at visible wavelengths after heat treatment can be easily predicted through the author's previous paper. Rather, it is necessary to present the difference in reflection behavior of 3D photonic crystals depending on whether or not there is a transfer process. That is, the advantage of uniform shrinkage in the vertical direction should be intensively discussed from an optical point of view.

Response 4:

Thank you for your helpful suggestion. We compared the bright-field reflective optical images of the post-processed woodpiles obtained with and without a transfer process. We found that conventional direct heating without a transfer process led to the deterioration of the color purity and structural stability of the woodpile. Therefore, the proposed method is advantageous for achieving uniform shrinking. We have added sentences to the manuscript to support our findings (page 15, lines 302–306; text highlighted in yellow).

“Meanwhile, we found that the post-processed woodpile obtained by conventional direct heating without a transfer process, i.e., the structure in Figure 3 (b), exhibited poorer color purity and structural stability (Figure S9) than the woodpile obtained with a transfer process. Therefore, the proposed method is advantageous for uniform shrinking.”

We have also included additional data in the Supplementary Information (Figure S9).

Figure S9. Comparison of the post-processed woodpiles with and without the pick-and-place process. (a, b) SEM and bright-field reflective optical images of the post-processed woodpile obtained using conventional direct heating. (c, d) Corresponding images of the

post-processed woodpile obtained using the proposed pick-and-place process.

Comment 5:

One-by-one transfer of 3D micro-objects to a receiving substrate using microneedles may be an interesting demonstration, but it is very inefficient and lacks prospects in terms of application. In particular, since TPL is one of the fundamentally low-productivity processes, the follow-up process proposed in this study further aggravates this distrust. Can the authors speed up the transfer process or suggest suitable applications for their process?

Response 5:

We thank the reviewer for highlighting this vital point. As the reviewer mentions, one important aspect of TPL is its use as a prototyping technology. However, its high printing accuracy and resolution are unique and advantageous for producing 3D structures with user-defined micro- and nanoscale features. Granted that this process is not currently suitable for applications that require mass production, it is nonetheless potentially useful in specific applications where a small volume number and specialized performance is needed. In other words, TPL will be able to serve specific fields different from other additive manufacturing technologies. Recent commercial TPL system models have demonstrated good productivity and throughput, with a print speed of over 100 mm/s. Regarding the transfer process, we can imagine a high throughput process in which samples are first attached to an intermediate substrate, such as thermal tape or soft PDMS, before the sacrificial layer is dissolved. These structural arrays can then be transferred onto a receiving substrate with good registration accuracy, followed by heat shrinkage.

Reviewer 3:

General Comment:

The authors present in this manuscript a new method based on using poly(vinyl alcohol) (PVA) as a sacrificial layer between the substrate and the printed structure, to avoid the deformation caused by the strong attachment between the original substrate and the prints. Subsequently, the 3D structures on the receiving substrate were heated and shrunk without distortion to obtain a uniformly scaled version of the original shape.

The manuscript is very well written, and suitable for publication in Nature Communications after addressing the minor comments:

Response:

We appreciate the reviewer's careful review of our manuscript and positive comments. Our detailed revisions are listed below to address the reviewer's concerns.

Comment 1:

The process includes partial thermal decomposition, at high temperature under air. About 80% of the material decomposes at about 450C, which is not very common to most organic materials. Therefore, the chemical composition of the photoresist should be given.

Response 1:

The IP-Dip2 photoresist that we used in our study consists of 60–80% 2-(hydroxymethyl)-2-[[1-(1-oxoallyl)oxy]methyl]-1,3-propanediyl diacrylate. During the heating process, the formation of volatile species such as CO and CO₂ by carbonization causes a large volume loss (please see Response 2 for Reviewer 1). Unfortunately, the vendor advised us that information available to the public should be kept to a minimum in accordance with scientific needs. Therefore, we disclosed minimal information on composition of the IP-Dip2 photoresist (page 7, lines 151–152; text highlighted in yellow; Figure S4 caption in the Supplementary Information).

“The IP-Dip2 photoresist is composed mainly of pentaerythritol triacrylate, which contains a large fraction of bonded oxygen.”

*“**Figure S4.** Raman spectrum of the IP-Dip2 photoresist, which consists of 60–80% 2-(hydroxymethyl)-2-[[1-(1-oxoallyl)oxy]methyl]-1,3-propanediyl diacrylate (CAS No. 3524-68-3, common name: pentaerythritol triacrylate).”*

Comment 2:

Page 8, figure 2: this is the first time laser power was mentioned. The contact and importance of these values should be explained in the manuscript before the figure capture.

Response 2:

Thank you for your recommendation. We have added sentences to address this issue (page 7, lines 132–134; text highlighted in yellow).

“All of the structures in Figure 2 were fabricated at a laser power of 27.5–30 mW to enable them to form with minimal shrinkage during the development process using an organic solvent.”

Comment 3:

Page 9, lines 174-177 "Lower laser power and heating temperature were used for the woodpile structure here than for the large 3D models in Figure 2 to avoid nanoweb formation between the woodpile lines during the printing process. These nanoweb are induced by the proximity effect and may greatly affect the optical performance (FigureS4)." Clear definitions are required for "nanoweb" and "proximity effect" as they are not adequately defined here and in Figure S4. The lack of clarity leads to a lack of understanding of the described.

Response 3:

Thank you for your comment, and we apologize for the unclear description in the previous version of our manuscript. When two lines are printed close to each other, a high density of radicals is generated between them during the exposure process; thus, narrow and discrete web-like structures (nanoweb) could spontaneously form between the lines, as seen in Figure S6 (i) and *Nano Futures* **2**, 025006 (2018).

Figure S6. As-printed woodpile structures prepared under different conditions. (a) Writing speed = 6 mm/s. (b) Writing speed = 9 mm/s. (d–i) Top view of SEM images corresponding to laser power = 20 mW (blue), 23 mW (green) and 28 mW (red) (xy (nominal) = 1.6 μm , writing speed = 9 mm/s). Scale bars: (d, f, h) 10 μm , and (e, g, h) 1 μm .

When the two lines are close enough, they may connect with each other or merge into a single unit. The proximity effect will induce laser damage when the excitation light intensity exceeds the energy required for the polymerization of the photoresist. As the laser power increases and the writing speed decreases, the proximity effect becomes more significant; thus, more nanowebbs are formed and laser damage is more likely to occur.

We have addressed this issue and added an explanation to the revised manuscript (page 10, lines 190–198; text highlighted in yellow).

“These nanowebbs are the narrow and discrete web-like structures that spontaneously form between two closely exposed structures induced by the proximity effect and may greatly affect the optical performance (Figure S6).^{20,28,45} When the two lines are close enough,

they may connect with each other or merge into a single unit. The proximity effect will induce laser damage when the excitation light intensity exceeds the energy required for the polymerization of the photoresist. As the laser power increases and the writing speed decreases, the proximity effect becomes more significant; thus, more nanowebs are formed and laser damage is more likely to occur.”

Comment 4:

Page 9, line 194, "only a tiny fraction of the surfaces came in contact with each other" Calculating the (normalized) contact area is recommended instead of relying on the particular description. This will provide a more accurate representation of the differences.

Response 4:

Thank you for highlighting this deficiency in our manuscript. Indeed, this statement is a critical point for us. Dieterich et al. and Sahli et al. directly observed the true contact area using a specific optical configuration (*PAGEOPH* **143**, 283–302 (1994) and *Proc. Natl Acad. Sci. U. S. A.* **115**, 471–476 (2018)). These authors discussed the relationship between friction and true contact area in detail. Unfortunately, because our structures are complex, not entirely rigid, and small, determining their true contact area is highly challenging. Therefore, for now, we have proposed only one of the plausible mechanisms in our process based on these authors’ work. Moreover, in this study, we focused on proposing a new process for uniform shrinking, applying it to complex 3D structures, and investigating their optical properties. The relevant mechanism should be investigated in the future and reported as a separate work. We have revised the sentence cited by the reviewer accordingly and added a reference to support our statement (page 11, lines 212–214; text highlighted in yellow).

“After the structures were detached from the original substrate and transferred to the receiving substrate, only a small fraction of the surfaces was assumed to have come into contact with each other.”

Additional Revisions

We have revised some technical terms in the following sentences.

Revision 1

Before

surface roughness and free energy of the receiving substrate

Corrected

the surface properties of the receiving substrate

page 2, line 36; text highlighted in yellow

Revision 2

Before

The surface roughness and surface free energy were measured to understand the uniform shrinking mechanism.

Corrected

The surface roughness, surface free energy, and adhesion force of the prepared structures were measured to understand the uniform shrinking mechanism.

page 4, line 91–93; text highlighted in yellow

Revision 3

Before

thermogravimetric and differential thermal analysis (TG-DTA)

Corrected

thermogravimetric analysis (TGA)

page 7, line 146; page 8, line 166; page 21, line 431; text highlighted in yellow

Revision 4

Before

Post-printed

Corrected
Post-processed

page 14, lines 277–279 in Figure 3 caption

Revision 5

Before

A 780 nm femtosecond pulsed laser was used and controlled with a laser power of 23–30 mW at a speed of 40 mm/s for 3D models (Figure 2) and 9 mm/s for small woodpile structures and 3D mascots (Figure 3, 4, and 5).

Corrected

A 780 nm femtosecond pulsed laser was used and controlled to a laser power of 27.5–30 mW at a speed of 40 mm/s for 3D models (Figure 2) and a laser power of 23–25.5 mW at a speed of 9 mm/s for small woodpile structures and 3D mascots (Figures 3, 4, and 5).

page 20, lines 397–400

REVIEWERS' COMMENTS

Reviewer #1 (Remarks to the Author):

In the revised manuscript, the authors have addressed the comments from the referees. Many experimental results, such as shrinkage uniformity and detailed thermogravimetric process, have been elucidated clearly. I would like to suggest its acceptance for publication on Nature Communications after the following minor problem.

In the reply to the 3th question of reviewer #1, the authors have explained that the effect of the surface roughness of PVA layer on the shrinkage result is negligible. The statement is still not be well supported. Please provide quantitative analysis about shrinkage rates of mounting different faces of the 3D-printed structures on the receiving substrate.

Reviewer #2 (Remarks to the Author):

The reviewer's responses are generally acceptable. The paper has obviously improved in quality through revision. I recommend that this paper be published in this journal.

Reviewer #3 (Remarks to the Author):

The authors have addressed my comments well (I read also the responses to the other reviewers too, and I think they responded properly). The revised paper is suitable for publication.

Reviewer Comments and Responses

Reviewer 1:

Comment 1:

In the revised manuscript, the authors have addressed the comments from the referees. Many experimental results, such as shrinkage uniformity and detailed thermogravimetric process, have been elucidated clearly. I would like to suggest its acceptance for publication on Nature Communications after the following minor problem.

In the reply to the 3th question of reviewer #1, the authors have explained that the effect of the surface roughness of PVA layer on the shrinkage result is negligible. The statement is still not well supported. Please provide quantitative analysis about shrinkage rates of mounting different faces of the 3D-printed structures on the receiving substrate.

Response 1:

We appreciate the reviewer's careful review of our manuscript and their recommendation of acceptance for publication in *Nature Communications* after we address a minor problem. The comments have helped significantly improve the quality of our manuscript. With respect to the reviewer's comment, we measured the shrinkage rates (SRs) of the 3D printed woodpile structures with different faces mounted on the receiving substrate using optical microscopy. As seen in the figure below, the obtained SR values were almost the same following the same shrinkage process described in the Methods section.

Supplementary Fig. 3 | Bright-field reflective optical images of the 3D printed structures with different faces mounted on the receiving substrate. a–e The as-printed

3D large woodpiles and **f–j** corresponding post-processed 3D large woodpiles. The *SR* of each structure was inserted in the corresponding image. The 3D large woodpiles were heated up to and maintained at 450 °C for 5 min. Scale bars = 20 μm.

According to the results, the *SRs* were not significantly affected by the different faces of the structures, that is, the effects of different surface roughness values are negligible. The main mechanism of our shrinkage process is that non-conformal contact between the printed structure and substrate. In the case of the as-printed structure mounted directly on glass, the contact is perfectly conformal. However, once detached, the contact is no longer conformal, which leads to reductions in the adhesion and ability of the printed structures to slide along the substrate surface during shrinking. The contact between different asperities is closely related to the shrinkage behavior. We have added the relevant discussion to the manuscript as well as an interesting article (Pastewka, L. & Robbins, M. O., Contact between rough surfaces and a criterion for macroscopic adhesion. *Proc. Natl Acad. Sci. U. S. A.* *III*, 3298–3303 (2014). 10.1073/pnas.1320846111) to the reference list so that readers can find more information on contact between rough surfaces (page 7, lines 144–146, and page 11, lines 212–213 and 225–231; text highlighted in yellow). We have also included the additional data in the Supplementary Information (Supplementary Fig. 3).

*“In addition, the *SRs* for the 3D printed structures with different faces remounted on the receiving substrate were mostly the same (Supplementary Fig. 3).”*

“The surfaces of the 3D printed structures and substrates generally exhibit nanoscale, microscale, and mesoscale asperities.”

“The main mechanism of our shrinkage process is that non-conformal contact between the 3D printed structure and substrate. In the case of the as-printed structure mounted directly on glass, the contact is perfectly conformal. However, once detached, the contact is no longer conformal, which leads to reductions in the adhesion and ability of the printed structures to slide along the substrate surface during shrinking. The non-formal contact between different asperities is closely related to the shrinkage behavior.”

Reviewer 2:

Comment 1:

The reviewer's responses are generally acceptable. The paper has obviously improved in quality through revision. I recommend that this paper be published in this journal.

Response 1:

We are grateful for the reviewer's careful review of our manuscript and recommendation for publication in the journal.

Reviewer 3:

Comment 1:

The authors have addressed my comments well (I read also the responses to the other reviewers too, and I think they responded properly). The revised paper is suitable for publication.

Response 1:

We appreciate the reviewer's thorough review of our manuscript and recommendation for publication in the journal.